# Extracts of *Talaromyces purpureogenus* Strains from *Apis mellifera* Bee Bread Inhibit the Growth of *Paenibacillus* spp. In Vitro

**DOI:** 10.3390/microorganisms11082067

**Published:** 2023-08-11

**Authors:** Katerina Vocadlova, Tim Lüddecke, Maria A. Patras, Michael Marner, Christoph Hartwig, Karel Benes, Vladimir Matha, Petr Mraz, Till F. Schäberle, Andreas Vilcinskas

**Affiliations:** 1Fraunhofer Institute for Molecular Biology and Applied Ecology (IME), Branch for Bioresources, Ohlebergsweg 12, 35392 Giessen, Germany; katerina.vocadlova@ime.fraunhofer.de (K.V.);; 2LOEWE Centre for Translational Biodiversity Genomics (LOEWE-TBG), 60325 Frankfurt, Germany; 3OncoRa s.r.o., Nemanicka 2722, 37001 Ceske Budejovice, Czech Republic; 4Retorta s.r.o., Tresnova 316, 37382 Borsov nad Vltavou, Czech Republic; 5Faculty of Agriculture and Technology, University of South Bohemia in Ceske Budejovice, Studentska 1668, 37005 Ceske Budejovice, Czech Republic; 6Institute for Insect Biotechnology, Justus-Liebig-University Giessen, Heinrich-Buff-Ring 26–32, 35392 Giessen, Germany

**Keywords:** *Apis mellifera*, honey bee, fungi, bee bread, *Talaromyces*, antimicrobial activity, biocontrol, natural product

## Abstract

Honey bees coexist with fungi that colonize hive surfaces and pollen. Some of these fungi are opportunistic pathogens, but many are beneficial species that produce antimicrobial compounds for pollen conservation and the regulation of pathogen populations. In this study, we tested the in vitro antimicrobial activity of *Talaromyces purpureogenus* strains isolated from bee bread against *Paenibacillus alvei* (associated with European foulbrood disease) and three *Aspergillus* species that cause stonebrood disease. We found that methanol extracts of *T. purpureogenus* strains B18 and B195 inhibited the growth of *P. alvei* at a concentration of 0.39 mg/mL. Bioactivity-guided dereplication revealed that the activity of the crude extracts correlated with the presence of diketopiperazines, a siderophore, and three unknown compounds. We propose that non-pathogenic fungi such as *Talaromyces* spp. and their metabolites in bee bread could be an important requirement to prevent disease. Agricultural practices involving the use of fungicides can disrupt the fungal community and thus negatively affect the health of bee colonies.

## 1. Introduction

The reproduction of most flowering plants and ~30% of all crops is dependent on pollination [1,2]. The estimated value of pollinators to the agricultural economy was USD 164 billion in 2009 [3] and now stands somewhere between USD 235 and USD 577 billion annually [4,5]. This value increases in line with crop production, and so does our dependence on pollinators for the maintenance of food security [2,4,6,7]. However, there has been a recent, dramatic decline in natural pollinator abundance and diversity [8,9]. Accordingly, managed pollinators such as honey bees (*Apis mellifera*, Linné 1758) (Hymenoptera: Apidae) play a key role in agroecosystems by augmenting wild pollinators and thus ensuring crop quality and yield stability. However, an array of biotic and abiotic stressors associated with agricultural intensification, parasites, and pathogens negatively affect the health of managed bees [10,11,12]. The deterioration in bee health results in the loss of colonies, particularly in the northern hemisphere [13,14,15]. Optimizing the health of individual bees is necessary to improve colony fitness, correlating directly with their benefits to humans, including pollination services as well as the production of honey.

Microorganisms that colonize animals (topically and internally) and their environments often fulfill essential functions [16]. The microbial communities associated with honey bees influence host metabolism, health, and stress tolerance [17,18,19,20,21,22]. Research has focused on the conserved core gut microbiome, which is dominated by nine clusters of bacterial species that develop 4–6 days after enclosure [23] and are present in almost all individuals regardless of geographic location or season [24,25,26,27]. In contrast, less attention has been paid to the composition and function of in-hive microbial communities that colonize the bee cuticle, hive surfaces, and food stores [28]. These consist of bacteria, yeast, and several genera of filamentous fungi that form the core mycobiome of pollen and bee bread, including *Aspergillus*, *Cladosporium, Botrytis*, *Penicillium*, *Alternaria, Mucor,* and *Rhizopus* [29]. Fungi are important for bee nutrition because they participate in pollen fermentation and its transformation into bee bread [30]. They also break down toxins [31] and confer resistance to fungal pathogens such as *Ascosphaera apis* (Maassen ex Claussen, L.S. Olive, and Spiltoir, 1955), the cause of chalkbrood disease, by producing antifungal compounds [32,33,34]. Some strains also reduce the viral load in the colony [35,36] and act as probiotics to promote the growth of symbiotic bacteria [37].

The genus *Talaromyces* (Benjamin, 1955) (Eurotiales: Trichomaceae) is a source of natural compounds with applications in medicine and the food industry (as producers of *Monascus*-like pigments) [38,39,40,41]. These fungi are known to antagonize plant pathogens and to be associated with insects [42]. The interactions within such complex ecological relationships are mediated by specialized microbial metabolites, and *Talaromyces* species produce diverse natural products that can be used for biological control [38,39].

Only a few studies have reported the presence of *Talaromyces* spp. associated with bees [34,36,42,43]. Here, we characterized *Talaromyces* strains recently isolated from honey bee bread [36] using a combination of colony morphology, DNA barcoding, and phylogenetic analysis. Bee-related *Talaromyces* strains have already been shown to inhibit human pathogens (*T. versatilis*) [43], mammalian and bee viruses (*T. purpureogenus*) [36], and fungal bee pathogens (*T. scorteus* and *T. dendriticus*) [34]. We tested the activity of organic crude extracts against the *Aspergillus* species (Eurotiales: Aspergillaceae) that cause stonebrood disease [44] and the bacterial opportunistic pathogen *Paenibacillus alvei* (Bacillales: Paenibacillaceae). This spore-forming bacterium is often isolated from colonies affected by European foulbrood disease (EFB), which is caused primarily by *Melissococcus plutonius* [45,46]. Both *Aspergillus* and *P. alvei* are also present in healthy colonies [47,48], but stonebrood disease is prevented by hygienic beekeeping practices and EFB has, until recently, been prevented by the use of oxytetracycline [49]. This is becoming less effective due to the ban on apiary antibiotics in many countries and the emergence of antibiotic-resistant pathogens, leading to heavy EFB infestations that often require the destruction of the colony to avoid disease spread [50]. The same measures might be needed for colonies severely infected with stonebrood disease, due to the health risk for beekeepers and consumers [51]. Defining the function of bee bread fungi such as *Talaromyces* strains and their metabolites in the control of these diseases would facilitate the development of new and more effective honey bee health protection strategies.

## 2. Materials and Methods

### 2.1. Bee Bread Collection and Fungal Cultivation

The fungal strains were isolated from *A. mellifera* bee bread collected in Kamenny Malikov, Czech Republic (KM Zirovnice; 49°12′51.533′′ N, 15°7′5.129′′ E) in March/April 2019 as previously described [36]. Briefly, hive frames containing stored pollen of various ages were cut out, and bee bread from 10 randomly selected cells was suspended in 0.9% NaCl containing 1% Tween-80 (Sigma-Aldrich, St. Louis, MO, USA). The suspension was then inoculated onto potato dextrose agar (PDA, VWR International, Radnor, PA, USA) at 25 °C. Fungi were subcultured several times until axenic isolates were obtained and identified. The *Talaromyces* strains were deposited in the Fraunhofer strain collection (EXT111748–EXT111754); other fungal genera were excluded from this study. To obtain spore suspensions, pure colonies were washed with 2 mL of 0.2% agar (Carl Roth, Karlsruhe, Germany) containing 0.05% Tween-80 [52], filtered through three layers of miracloth and stored at 4 °C. Three 1-µL drops were inoculated onto solid media: Czapek yeast autolysate (CYA) medium, malt extract agar (MEA), yeast extract supplemented (YES) medium, creatine sucrose agar (CREA) (according to [52]), and Sabouraud’s dextrose agar (SDA)—30 g/L Sabouraud dextrose broth (Merck Millipore, Burlington, MA, USA) and 15 g/L agar (Carl Roth, Karlsruhe, Germany). After 7 or 10 days, colonies were transferred to liquid malt peptone (MP) medium—30 g/L malt extract (Thermo Fisher Scientific, Dreieich, Germany), 5 g/L mycological peptone from meat (Carl Roth, Karlsruhe, Germany) as previously described [36]. The workflow is summarized in Figure 1. Images of fungal colonies were captured using an EOS 450D camera (Canon, Tokyo, Japan) and edited in GIMP ver. 2.10.34 [53].

### 2.2. Extract Preparation for Chemical Analysis and Antimicrobial Tests

Extracts were prepared from cultures grown on solid CYA and YES medium for 10 days [52]. Briefly, three plugs were cut from the center of single colonies and transferred to 2 mL screw cap vials. We added 0.8 mL ethyl acetate containing 1% formic acid and incubated in an ultrasonic bath (35 kHz) for 30 min [52]. The liquid phase was transferred to a fresh vial and evaporated under nitrogen, and the residues were redissolved in 40 µL methanol and stored at 4 °C. The fungal strains were also incubated in liquid MP medium and were lyophilized and extracted in methanol as previously described [36]. The crude extracts were redissolved in methanol to a concentration of 100 mg/mL and were stored at –20 °C. Before UHPLC-HR-MS analysis, all samples were centrifuged (8000× *g*, 1 min, room temperature), and 30–40 µL was transferred to 2-mL HPLC vials with glass micro-inserts.

### 2.3. Molecular Barcoding and Phylogenetics

We collected mycelia from 7-day-old colonies grown on SDA using a sterile scalpel and homogenized the tissue under liquid nitrogen with a mortar and pestle. The ground tissue (50–80 mg) was mixed with 490 µL 2 × CTAB buffer (2% CTAB, 1.4 M NaCl, 20 mM EDTA, 100 mM Tris of pH 8) preheated to 60 °C. We then added 10 µL of proteinase K (50 mg/mL) and incubated at 60 °C for 1 h on a shaking platform. The lysate was extracted by adding 500 µL 24:1 chloroform:isoamylalcohol, and the aqueous layer (400 µL) containing was transferred to a new 1.5 mL Eppendorf tube. The DNA was precipitated by adding 270 µL isopropanol. After centrifugation (14,000× *g*, 10 min, room temperature), the pellet was washed with 1 mL ice-cold 70% ethanol and dried in a vacuum concentrator for 10 min. The DNA was dissolved in 20 µL of TE buffer with 2 µL of RNase A (10 mg/mL) for 3 h at 37 °C. Samples were then stored at –20 °C. The concentration and quality of DNA was determined using a NanoDrop UV spectrophotometer and by 1% agarose gel electrophoresis.

To create the barcodes, we amplified four DNA regions: β-tubulin (BenA), calmodulin (CaM), the internal transcribed spacer (ITS), and RNA polymerase II second largest subunit (RBPII), which are often used for the molecular identification and phylogenetic analysis of fungal isolates [52,54] (Table 1). Each 20 µL PCR mix contained 10 µL Taq 2× Master Mix, 0.25 µM of each primer, 100–200 ng of template DNA, and PCR-grade water to top up. The reaction conditions are summarized in Table 2. PCR products were purified using ExoSAP-IT (Thermo Fisher Scientific, Waltham, MA, USA) according to the manufacturer’s instructions but using modified thermal conditions (37 °C for 35 min, 85 °C for 15 min, and 10 °C infinite hold). The cleaned PCR products were sent with forward and reverse primers for bidirectional sequencing (Eurofins, Val Fleuri, Luxembourg).

Sequence reads were analyzed and assembled using Geneious v10.2.6, and the assembly consensus sequences of the barcodes were compared to the sequences in the International Sequence Database (INSD) using the GenBank basic local alignment search tool (BLAST) [60]. The search was adjusted for comparison with the type and reference material and the RefSeq Targeted Loci database (available only for ITS sequences). For species assignment based on ITS sequences, we used the recommended identity threshold of ≥97% [61]. Due to the lack of available Ben1A sequences from the type material in the database (December 2022), phylogenetic analysis was carried using the Ben2A, CaM, ITS, and RBPII sequences (Appendix A).

Sequence alignments were prepared using MUSCLE in Geneious v10.2.6, and single gene trees (Appendix A) were constructed using IQtree with default settings [62,63]. From the concatenated alignment, uninformative sites were removed using Gblocks 0.91b [64,65]. Next, we used partition finder in Cipres Science Gateway v3.3 [66] to determine best-fitting models of molecular evolution and partition schemes. Two simultaneous phylogenetic analyses were carried out using MrBayes v3.2 [67,68,69] with four Markov chains (three heated, one cold) for 10 million generations with a sampling frequency of 1000 and a 25% burn-in. From the resulting trees, a 65% majority-rule consensus tree was constructed and visualized using iTol v6 [70].

### 2.4. Detection of Talaromyces spp. at Different Time Points and Locations

Samples were collected in July 2019 and April and July 2020 from five apiaries in the South Bohemia region of Czech Republic as described in Section 2.1. The collection sites were Kamenny Malikov village (KM Zirovnice and KM agro) and three apiaries in Ceske Budejovice (CB campus, CB Litvinovice, and CB Kroclov) from urban and suburban areas and the city periphery, respectively. The locations were ranked by anthropogenic influence, focusing on the level of urbanization and the agricultural landscape within ~6 km (Appendix A). ITS sequences were used for species assignment.

### 2.5. UHPLC-HR-MS Analysis and Metabolic Fingerprinting

Samples were processed for metabolic fingerprinting as previously described [71]. Briefly, the samples were fractionated on a 1290 UHPLC system (Agilent, Santa Clara, CA, USA) equipped with DAD, ELSD, and a maXis II (Bruker, Billerica, MA, USA) ESI-qTOF-HRMS. We used a gradient of 0.1% formic acid in water (buffer A) and 0.1% formic acid in acetonitrile (buffer B) at a flow rate of 600 µL/min. The gradient began at 95% A and was held for 0.30 min before a transition to 4.75% A over 18.00 min and 0% A over 18.10 min, with a hold for 22.50 min. The gradient then increased to 95% A over 22.60 min followed by a hold for 25.00 min. The column oven temperature was set at 45 °C, and we used an Acquity UPLC BEH C18 1.7 µm column (2.1 × 100 mm) with an Acquity UPLC BEH C18 1.7 µm VanGuard pre-column (2.1 × 5 mm).

For micro-fractionation [42], extracts were injected into the UHPLC-HR-MS system described above. However, the flow path was changed so that 90% of the flow was diverted to a custom-made fraction collector (Zinsser–Analytik, Frankfurt, Germany) while the rest was analyzed in MS/MS mode in the maXis II. Collision-induced fragmentation at 28.0–35.05 eV was achieved using argon at 10^−2^ mbar. Micro-fractionation assay plates were prepared by injecting 2 or 5 µL of extract. For each injection, 159 fractions were generated and collected on a 384-well plate (fraction length: 7 s). Before screening, the plates were dried under a vacuum using an HT12-II centrifugal concentrator (Genevac, Ipswich, UK) at 35 °C.

For metabolic fingerprinting, raw MS data were processed using DataAnalysis v5.3 (Bruker) including recalibration with sodium formate, followed by RecalculateLinespectra with a threshold of 10,000 and subsequent FindMolecularFeatures (0.5–25 min, S/N = 0, minimal compound length = 8 spectra, smoothing width = 2, correlation coefficient threshold = 0.7). Bucketing was achieved using ProfileAnalysis v2.3 (30–1080 s, 100–6000 *m*/*z*, advanced bucketing for 12 s at 5 ppm, no transformation, bucketing basis = H+). We then constructed a cosine similarity heat map.

### 2.6. Evaluation of Antimicrobial Activity

#### 2.6.1. Antifungal Activity Test

Antifungal activity was tested against three *Aspergillus* species associated with stonebrood disease (*A. flavus* ATCC9170, *A. fumigatus* ATCC10894, and *A. niger* ATCC10549) from Fraunhofer strain collection (STO20519-STO29521) using CLSI M51-A parameters [72]. The *Aspergillus* strains were cultivated on PDA (VWR International, Radnor, PA, USA) for 7 days at 25 °C. The spores were washed by 5–6 mL of 0.05% Tween-80 (Sigma-Aldrich, St. Louis, MO, USA) and filtered through three layers of miracloth to remove the hyphal structures. The inoculum was adjusted to OD_530_ = 0.14–0.46 based on spore morphology [73]. The inoculum was spread evenly over the surface of Mueller–Hinton agar (Carl Roth, Karlsruhe, Germany) using a sterile cotton swab. The depth of the medium was consistent in each Petri dish (4 mm). The extracts from the liquid *T. purpureogenus* strains cultures (Figure 1) were diluted with methanol to 10 mg/mL, and 25 µL of the crude extract was applied to 6 mm cellulose disks and left in the laminar flow cabinet for 30 min to dry. Methanol was used as a control. The disks were placed on the inoculated media and pressed down with sterile forceps. The zones of inhibition were evaluated in triplicate after incubation at 35 °C for 24 h.

#### 2.6.2. Antibacterial Activity Test

We tested the antibacterial activity of the extracts in a dilution series (25–0.39 mg/mL) in triplicate using a micro-broth dilution test against *P. alvei* CCM 2051 provided by the Czech Collection of Microorganisms (Masaryk University, Brno, Czech Republic) as previously described [74] with modifications. Briefly, extracts were diluted 1:1 with Mueller–Hinton broth, yeast extract, potassium phosphate, glucose, and pyruvate (MYPGP), and 100 µL of the solution was pipetted to the first row of the microdilution 96-well plate. A two-fold dilution series was prepared by transferring the extract solutions (50 µL) to the next rows containing 50 µL of the MYPGP. Dilutions of methanol were included as a solvent growth control. The lyophilized bacteria were cultivated at 37 °C for ~24 or ~48 h on MYPGP agar. The inoculum was diluted in sterile water and adjusted to 0.5 McFarland units using the DEN-1 McFarland densitometer (BioSan, Riga, Latvia). The bacterial suspension was diluted 1:150 with the MYPGP (~10^6^ CFU/mL), and 50 µL was added to each well within 15 min. Bacterial growth was measured at 625 nm using an xMark spectrophotometer (Bio-Rad, Hercules, CA, USA). Minimum inhibitory concentrations (MICs) were defined as the lowest concentration of extract or standard that inhibited growth by at least 80% relative to the control (bacterial suspension with no extract/solvent).

#### 2.6.3. Bioactivity-Guided Dereplication

The potency of the *T. purpureogenus* strains extracts was determined using micro-broth dilution assays as previously described [75,76,77]. Briefly, extracts were screened in a 12-point dilution series (2–0.001 mg/mL) in triplicate. In addition to four *Paenibacillus* strains (*P. lautus* DSM3035, *P. lactis* FH1832, and two unspecified taxa from the Fraunhofer strain collection ST133196 and ST514408 [78]), we also included the control strain *Escherichia coli* ATCC35218 and the type strain *Bacillus subtilis* DSM 10. For all bacteria, the density of the overnight pre-cultures, incubated in cation-adjusted Mueller–Hinton II medium (BD) at 37 °C while shaking at 180 rpm, was adjusted to 5 × 10^5^ cells/mL. The assay plates were set up by pipetting 100 µL of adjusted bacterial suspension in each well, except the medium control wells H01-H05 (blank or “low”). Wells H06-H12 contained bacterial suspensions without extracts or antibiotics (growth control, “high”). On each assay plate, we also assessed the MIC of three reference antibiotics (ciprofloxacin 0.5–0.0002 µg/mL, cefotaxime, and gentamicin both 64–0.03 µg/mL). The dilution series of pure MeOH, the sample triplicate solved in MeOH, and the reference antibiotics was prepared by adding additional 98 µL of bacterial suspension plus 2 µL of the respective test samples to the first well of rows A–G. Hence, the maximal solvent concentration per well was 1%. Next, the 1:2 dilution series were prepared by transferring 100 µL suspension. After the last dilution step in column 12, the remaining 100 µL were discarded. Assay plates were incubated for 18 h at 37 °C and 85% relative humidity, shaking at 180 rpm, before we measured the turbidity at 600 nm on a LUMIstar Omega microplate spectrophotometer (BMG Labtech, Ortenberg, Germany) as a proxy for cell growth. The relative growth inhibition was calculated based on the absorbance units (AU) of the sample and the controls (see above) using the following formula:$$\mathrm{rel}.\;\mathrm{inh}.\%=100\;*\;\left[1-\frac{\mathrm{AU}\;\mathrm{sample}\;-\;\mathrm{AU}\;\mathrm{Low}}{\mathrm{AU}\;\mathrm{High}\;-\;\mathrm{AU}\;\mathrm{Low}}\right]$$

MICs were defined as above. Extracts with bioactivity observed over at least three dilution steps (≤0.5 mg/mL) were selected for micro-fractionation. Selected extracts were injected twice into our UHPLC-HRMS-MS system (2 and 5 µL µL, see Section 2.5). The dried assay plate was re-screened against *P. lautus* by adding 50 µL of adjusted pre-culture (see above) to each well except column 1 (media blank). Columns 2 and 3 contained no fractions, but a dilution series of gentamicin ranging from 256–0.008 µg/mL. Column 4 contained only bacterial suspension (growth control), while the remaining 320 wells contained 2 × 159 fractions collected from the two injection and an unfractionated extract control of the respective volume. Incubation, read-out, and calculation of the relative growth inhibition was carried out as described above.

## 3. Results

### 3.1. Colony Morphology

The use of four standard media enabled us to distinguish the fungal strains based on specific morphological traits: size, sporulation color, and pigment production (Figure 2). The size of the cultures on MEA was comparable (Table 3) but differences in red pigment production (*Monascus*-like red pigments) were observed in the fruiting body (B18 > B13 > B195 on reversed MEA and YES) and media (B13 > B195 > B18 on CYA). The production of low levels of acid on CREA was observed only for strain B49 (Figure 2, Table 3).

Our cultivation-dependent method detected isolates of the genus *Talaromyces* in bee bread from three of the five apiaries, but only in a few hives in spring (Table 4). Based on ITS sequencing, we identified *T. purpureogenus* in apiaries in areas with a low anthropogenic influence (CB Kroclov and KM Zirovnice) and two species (*T. purpureogenus* and *T. piceae*) in urban samples (CB campus). We did not detect any *Talaromyces* spp. in the other apiaries (KM agro and CB Litvinovice). Images of the isolates are provided in Appendix A. Fungal isolates representing other genera were not considered.

### 3.2. Molecular and Phylogenetic Characterization

We assigned all strains to the species *T. purpureogenus* based on ITS sequence similarity, thus excluding other species in the RefSeq ITS database (Table 5). The sequences were uploaded to the NCBI database (OR192894–OR192900, Appendix A). By comparing these sequences with the database, we also observed the highest identity (>98%) with *T. purpureogenus* for the other three barcodes (Ben2A, CMD, and RBPII). Only one Ben1A sequence from *Talaromyces* spp. type material was available in the database—*T. stipitatus* (query cover/identity (QC/I) = 88/96.45%; XM_002341495.1). In the standard database, the Ben1A sequences (OR327661–OR327667) showed the highest QC/I to *T. marneffei* (>99/>93.48%; CP045656.1) and less than 90% identity to *T. purpureogenus* isolated from medicinal plants (QC/I = >95/89.65–89.82; HM596783.1) [79].

The resulting phylogeny supports the monophyletic nature of *Talaromyces* and is composed of several subclades containing different members of this genus (Figure 3). General support for the *Talaromyces* clade was high, but several of the deepest clades were only weakly supported (see Appendix A for numeric bootstrap values). However, the overwhelming majority of shallow clades relevant for the taxonomic placement of our isolates were sufficiently supported to draw firm conclusions.

The most ancestral clade within our phylogeny contained two taxa (*T. aurantiacus* and *T. argentinensis*) and was followed by a three-taxon clade (*T. veerkampii*, *T. louisianensis,* and *T. caifornicus*). The next clade contained four taxa (*T. malicola*, *T. tumui*, *T. pratensis,* and *T. domesticus*) with others splitting off subsequently. The first taxon to split off was *T. aculeatus*, followed by *T. sparsus* and *T. panamensis*, then a clade containing *T. aspriconidius*, *T. haitouensis,* and *T. flavus*, and another with *T. francoe*, *T. thalandensis,* and *T. penicilloides*. The remaining taxa included in our analysis formed the two youngest clades. The first contained two taxa (*T. zhenhaiensis* and *T. stipitatus*) and was placed as a sister to a clade containing all our isolates and different strains of *T. pupurogenus* (CBS 184.27, CBS 113158, CBS 122434, CBS 108923, CBS 286.36, CBS 113161, and DTO189A1). This larger clade, containing the *T. zhenaiensis*/*T. stipitatus* clade plus the *T. purpureogenus* clade, received substantial support (99%). This also applied to both included subclades, which received 100% support for their monophyletic composition. Similar topologies of our isolates in close proximity to *T. purpureogenus* were consistently recovered across all single-gene trees (Appendix A). Overall, our phylogenetic analysis supports the assignment of our isolated fungi to the species *T. purpureogenus* as suggested by our ITS DNA barcoding data (Table 5).

### 3.3. Metabolic Fingerprinting

The extracts showed a broad distribution of compounds over the polarity range (Appendix A). The pairwise cosine similarities of all samples were plotted as a heat map (Figure 4). The main grouping correlated with the solid and liquid media, as expected. The clustering order in the solid media was less influenced by the medium (YES and CYA) and the metabolic features of the solid medium controls were therefore very similar (cosine 0.93). The medium controls and strains B11, B13, B30, and B195 clustered in pairs (consisting of one sample each from CYA and YES). Strains B49 and B69 were merged into one cluster. Strains B18 and B195 clustered together in YES, but the metabolic profile of B18 on CYA was distinct from the other strains in this cluster.

In liquid media, the highest similarity in metabolic features was observed between two pairs – strains B30 and B49 (0.91) and strains B11 and B69 (0.83). Whereas all five strains (B11, B13, B30, B49, and B69) were merged into one cluster, strains B18 and B195 were clustered more closely (0.60) and the metabolic features were more distant from those of the other strains. The clustering order of the strains and the corresponding pairwise cosine similarities are summarized in Appendix A.

### 3.4. Antimicrobial Activity and Bioactivity-Guided Dereplication

The crude extracts showed no ability to inhibit the growth of any of the *Aspergillus* species (no zone of inhibition) at a concentration of 0.25 mg of per disc. Extracts B195 and B18 inhibited the growth of *P. alvei* at an MIC of 0.39 mg/mL, whereas extracts B11 and B69 showed activity only at higher concentrations, with MICs of 6.25 and 3.13 mg/mL, respectively (Table 6).

To support the results of the antimicrobial activity against the bee pathogen *P. alvei*, we expanded the screening panel to include further *Paenibacillus* species as well as the quality-control strain *E. coli* ATCC35218 and type strain *B. subtilis* DSM 10. Interestingly, extract B18 inhibited the growth of *P. lautus* and *P. lactis* at MIC = 0.5 mg/mL (Table 6). The previously observed ability of extract B195 to inhibit *P*. *alvei* was also observed against *P. lautus* and *P. lactis,* but only at the highest tested concentration (MIC = 2 mg/mL). Extract B195 also displayed moderate activity against *B. subtilis* at 2 mg/mL. None of the extracts inhibited *E. coli* at the concentrations we tested. The full dataset is provided in Appendix A. Based on the above results, extract B18 was used for micro-fractionation and re-screened against *P. lautus*, for which optimal growth conditions have been established in our laboratory.

The B18 extract yielded five zones of consecutive fractions that inhibited the growth of the test strain, namely fractions 26–28, 30–31, 33–34, 39, and 114 (Table 7). The fractionation, MS, and MS/MS spectra are provided in Appendix A. Dereplication was achieved by aligning the major ions in active fractions to commercial and in-house MS/MS reference databases. The two main signals in the average mass spectrum of fractions 26–28 were the single protonated ions at *m*/*z* 245.1283 [M+H]⁺ and *m*/*z* 211.1438 [M+H]⁺, corresponding to the molecular formulae C_14_H_16_N_2_O_2_ and C_11_H_18_N_2_O_2_, respectively. The compounds were identified as the diketopiperazines cyclo-Phenyl- alanyl-Prolyl and cyclo-Leucyl-Prolyl (Figure 5: 1–2). Fractions 30–31 contained a group of stereoisomers of cyclo-Phenylalanyl-Prolyl (Figure 5: 1) as well as *m*/*z* 543.2176 [M+H]⁺, which was assigned the molecular formula C_24_H_34_N_2_O_12_. Interestingly, this compound did not match any reference compound in our internal MS/MS database, and a molecular formula search in commercial natural product databases (DNP, Dictionary of Natural Products and AntiBase) also retrieved no hits. The identity of the compound remains unknown. An isomer of the same compound was identified in fractions 33–34. In addition, these fractions contained 5,6,8-trihydroxy-3-methyl-2-benzopyran-1-one (C_10_H_8_O_5_, ionizing at *m*/*z* 191.0334 [M-H_2_O+H]⁺, *m*/*z* 209.0434 [M+H]⁺ and *m*/*z* 417.0807 [2M-H_2_O+H]⁺) (Figure 5: 3). Fraction 39 contained the siderophore l-ornithine, *N*^2^-acetyl-*N*^5^-hydroxy-*N*^5^-(5-hydroxy-3-methyl-1-oxo-2-pentenyl)-, trimol. ester, (*Z*,*Z*,*Z*)-(9CI) (C_39_H_62_N_6_O_16_, ionizing at *m*/*z* 436.5186 [M+2H]²⁺ and 871.4286 [M+H]⁺) (Figure 5: 4). A second major ion in fraction 39 (*m*/*z* 516.2204 [M+H]^+^) could not be dereplicated. Ultimately, fraction 114 contained one major ion (*m*/*z* 281.2482 [M+H]⁺) and a minor ion (*m*/*z* 379.3368 [M+H]⁺). Neither compound aligned with any reference compound spectrum. The predicted molecular formula search of the major ion (C_18_H_32_O_2_) retrieved >100 candidates in the DNP, including ubiquitous linolenic acid. However, the nonspecific fragmentation did not allow structural assignment.

## 4. Discussion

The role of non-pathogenic fungi as regulators of pathogens in the hive and in bee bread has been addressed in several studies [32,33,34]. Indeed, bee bread is colonized by many filamentous fungi whose potential as biological control agents is yet to be explored. Here, we characterized seven strains of *Talaromyces* recently found in honey bee bread and tested the antimicrobial activity of methanol extracts against bacteria associated with EFB (*P. alvei*) and opportunistic pathogens of the genus *Aspergillus*.

Based on morphology, DNA barcoding, and phylogenetic analysis, we assigned all seven strains to the species *T. purpureogenus* (Samson, Yilmaz, Frisvad, and Seifert 2011), formerly known as *Penicillium purpureogenum* (Stoll, 1904). *Talaromyces* species have previously been recovered from *A. mellifera* honey [80], *A. cerana* bee bread [34], dead adults of *A. dorsata* (Fabricius, 1793) [43], and stingless bees (*Melipona* spp.) [81]. However, to the best of our knowledge, the species *T. purpureogenus* had not been identified in honey bees before, although it has been found in Coleoptera (bark beetles), Diptera (mosquitoes), Hymenoptera (ants), Hemiptera, and Trichoptera [42].

Although few studies have reported the presence of *Talaromyces* in bee bread, its presence is not surprising because this ubiquitous genus can be collected by bees from various sources [54]. Bees passively or actively collect fungi during foraging and incorporate them into the corbicular pollen [82,83]. Interestingly, some fungi are probably introduced or eliminated by the bees during the collection and storage of pollen [30]. We isolated *Talaromyces* strains from some of the hives in one rural location, an urban area, and the city periphery. Seasonal factors may be relevant because we found the *Talaromyces* strains only in spring. Other studies exploring the composition of fungi in bee bread in Europe did not find any *Talaromyces* species, but samples were collected in the summer [29,84], when the absence of *Talaromyces* is consistent with our results. In worker bees, the prevalence of fungi was higher in fall and winter [32]. The composition of fungi in bee bread during months with no flight activity has not been studied.

We did not find any *Talaromyces* species in apiaries located directly on agricultural land. The foraging distance of *A. mellifera* ranges from several hundred meters to several kilometers, depending on the available pasture [85,86]. The presence of agricultural land (such as a canola field) within the foraging range increases the risk of fungicide residues in the food stores that influence the fungal community. The negative effect of agricultural pressure and fungicide contamination on the bee bread fungal community has been confirmed by sequence-based [29] and cultivation-based analysis [87]. Our results suggest that seasonal effects and agricultural pressure determine the presence of *Talaromyces* in bee bread. However, more research is needed to correlate the presence and seasonal prevalence of *T. purpureogenus* and other *Talaromyces* species in bee bread.

Our in vitro assays revealed that extracts of *Talaromyces* strains B18 and B195 showed antibacterial activity against the pathogen *P. alvei* and other tested strains of *P. lautus* and *P. lactis*. Bioactivity-guided dereplication in assays against *P. lautus* revealed multiple fractions and corresponding inhibitory compounds (siderophores, diketopiperazines, and three unknown compounds) that are also likely to show activity against the other *Paenibacillus* species. Siderophores are metal chelators that facilitate the uptake, intracellular transport, and storage of iron in plants, fungi, and bacteria, making them useful for both medical and environmental applications [88]. Most fungal siderophores are acylated hydroxamates [89], such as the coprogen-type talarazines produced by *Talaromyces* [90]. The biosynthesis of siderophores plays a key role in fungal virulence and influences their interactions with other microbes [91]. The ability to produce iron-chelating molecules can starve some microbes of this essential nutrient and thus cause growth suppression while providing a source of iron for others, such as yeasts, thus promoting their growth [91]. Several fungal compounds have been associated with regulation of the bee bread microbiome, including organic acids [34] and lovastatin [92]. Given that siderophores inhibited the growth of bacteria in our study, it would be interesting to explore the use of siderophores to regulate EFB and AFB (American foulbrood) in vivo. The unknown active compounds should also be investigated as novel anti-infectives.

Most sequence-based studies identify fungi to the genus or species level. Our results highlight the metabolic diversity of different strains of the species *T. purpureogenus*. Depending on the strain and cultivation conditions, *T. purpureogenus* isolated from the bee bread can affect colony health and pathogen resistance in different ways, such as the production of antimicrobial compounds as discussed here, as well as mycotoxins such as the rubratoxins found in our previous study [36]. The genus *Talaromyces* may consist of an assemblage of species/strains with various metabolic activities, as described for bee-related *Aspergillus* species [93]. These fungi are found in the bee hive as a mixture of atoxigenic and toxigenic strains, and some of them are opportunistic pathogens [93]. Similarly, strains of other genera associated with bee bread, such as *Penicillium* and *Alternaria* [94,95], are assumed to be beneficial but can produce both the antimicrobial compounds and mycotoxins that are lethal to bees. Therefore, the nature of the bee bread mycobiome depends on the balance between different fungal strains, which under ideal circumstances are beneficial to the health of bees and the colony as a whole.

## 5. Conclusions

Fungi in bee bread play an important role in the regulation of pathogens, conferring resistance and promoting colony survival. Many bee bread fungi, including *T. purpureogenus*, can produce protective antimicrobial compounds as well as lethal mycotoxins. The strain-level balance is therefore important for the beneficial function of the mycobiome. Anthropogenic activity, such as the use of fungicides in agriculture, can disrupt this balance and negatively affect colony health.

## Figures and Tables

**Figure 1 microorganisms-11-02067-f001:**
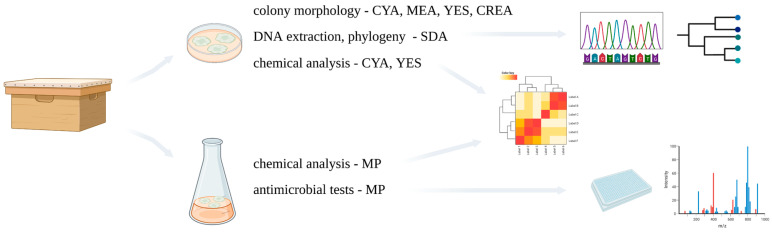
Workflow showing the use of cultivation media prior to colony morphology assessment, molecular barcoding, and phylogenetic analysis (top right), chemical analysis (cosine similarity, center), antimicrobial assays and bioactivity-guided dereplication (bottom right). Figure created with BioRender.com.

**Figure 2 microorganisms-11-02067-f002:**
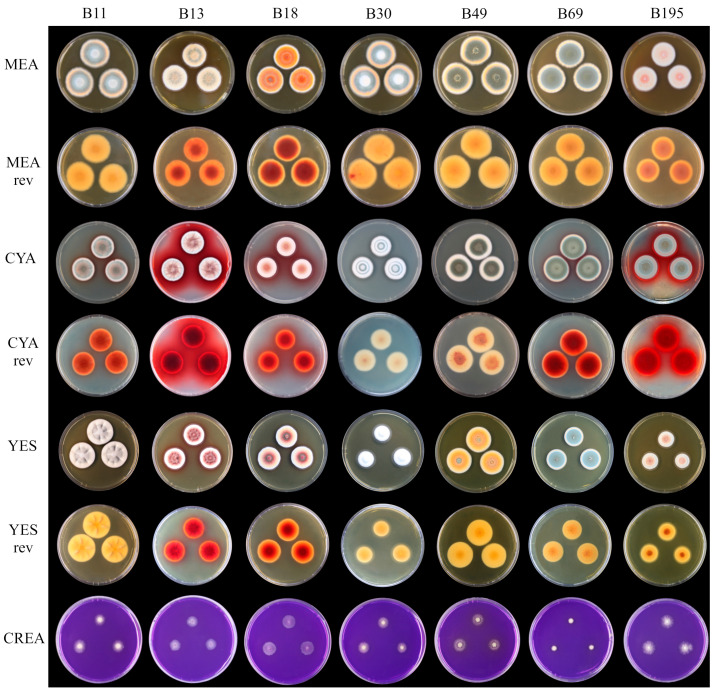
Colony morphology of *Talaromyces purpureogenus* strains isolated from bee bread (columns) when grown on different types of media (rows).

**Figure 3 microorganisms-11-02067-f003:**
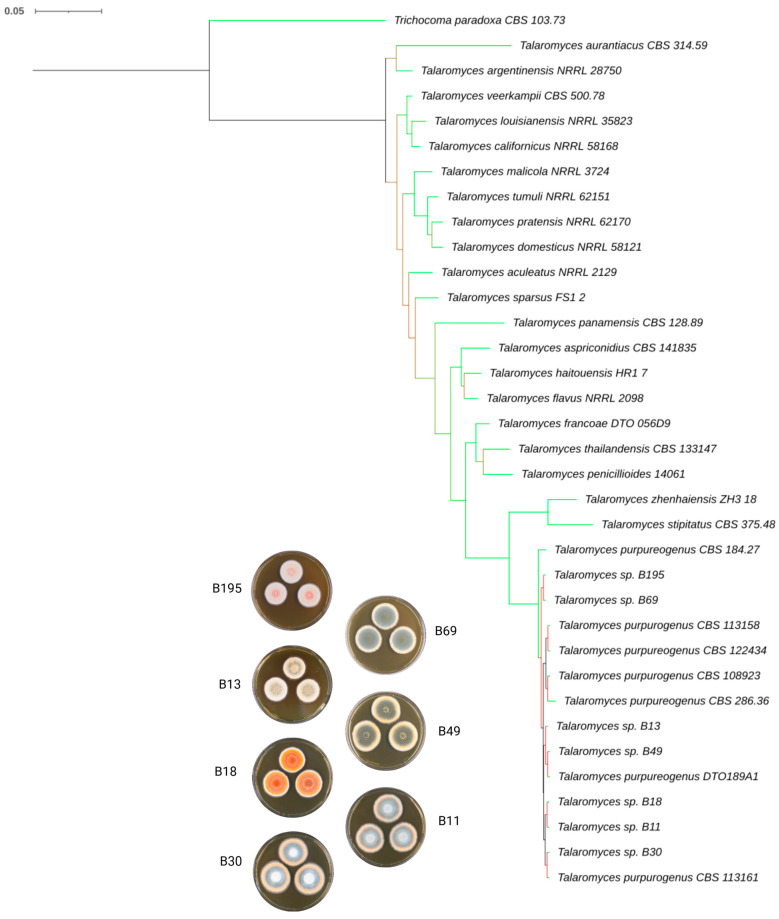
Phylogenetic tree based on ITS, Ben2A, CaM, and RBPII gene regions showing the relationship between the seven newly described *T. purpureogenus* strains and other members of the genus *Talaromyces*. The bootstrap values are indicated using a green–red color model (green = high support and red = low support). The numeric values can be found in Appendix A. *Trichocoma paradoxa* was used as the outgroup.

**Figure 4 microorganisms-11-02067-f004:**
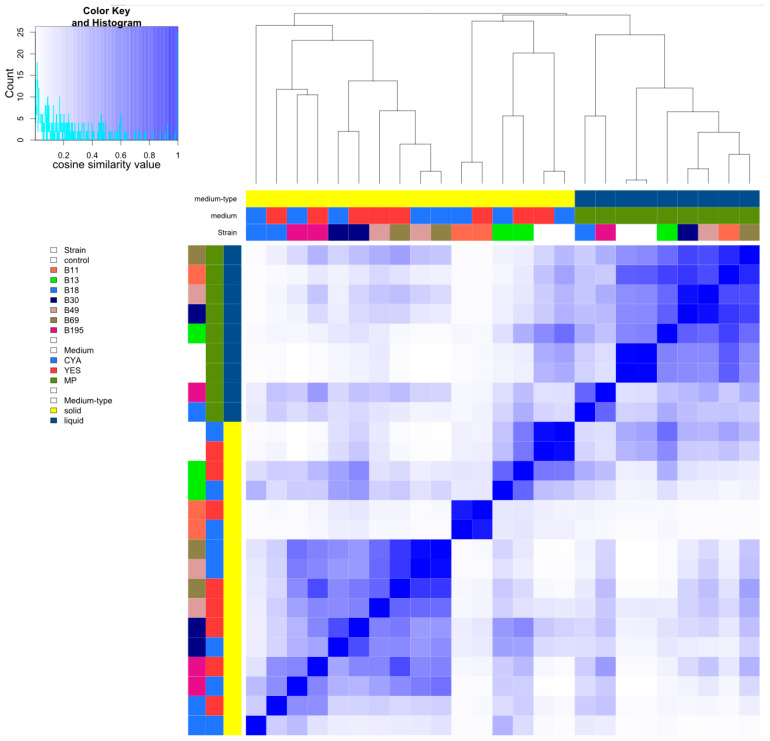
Cosine similarity heat map of extracts analyzed by mass spectrometry.

**Figure 5 microorganisms-11-02067-f005:**
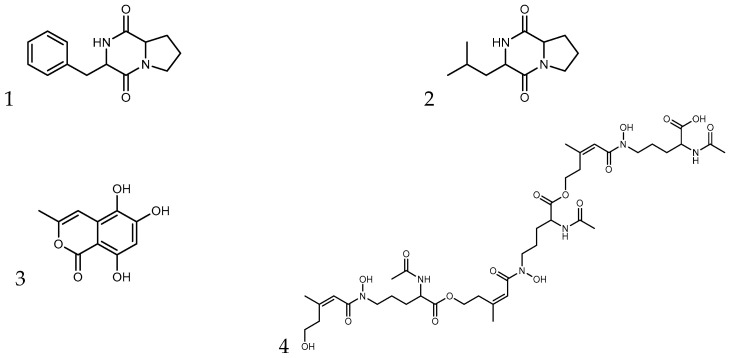
Chemical structures of the compounds identified in the active fractions. 1 and 2—diketopiperazines cyclo-(Phenylalanyl-Prolyl) and cyclo-(Leucyl-Prolyl), resp.; 3—5,6,8-trihydroxy-3-methyl-2-benzopyran-1-one; 4—siderophore l-ornithine, *N^2^*-acetyl-*N^5^*-hydroxy-*N^5^*-(5-hydroxy-3-methyl-1-oxo-2-pentenyl)-, trimol. ester, (Z,Z,Z)-(9CI).

**Table 1 microorganisms-11-02067-t001:** Primer pairs used for the amplification of the four gene regions.

Target	Name	Orientation	Sequence (5′→3′)	Reference
BenA	Bt2a	For	GGTAACCAAATCGGTGCTGCTTTC	[55]
	Bt2b	Rev	ACCCTCAGTGTAGTGACCCTTGGC	
	Bt1a	For	TTCCCCCGTCTCCACTTCTTCATG	
	Bt1b	Rev	GACGAGATCGTTCATGTTGAACTC	
ITS	ITS1F	For	CTTGGTCATTTAGAGGAAGTAA	[56]
	ITS4	Rev	TCCTCCGCTTATTGATATGC	[57]
CaM	CF1	For	GCCGACTCTTTGACYGARGAR	[58]
	CF4	Rev	TTTYTGCATCATRAGYTGGAC	
RBPII	5F2	For	GGGGWGAYCAGAAGAAGGC	[59]
	7CR	Rev	CCCATRGCTTGYTTRCCCAT	

**Table 2 microorganisms-11-02067-t002:** PCR thermal profiles for the used primers.

	ITS and CaM	Ben1A and Ben2A	RBPII
Initial denaturation	95 °C/5 min	94 °C/5 min	95 °C/5 min
Denaturation	94 °C/45 s	94 °C/45 s	94 °C/45 s
Annealing	55 °C/45 s	50 °C/45 s	52 °C/45 s
Extension	72 °C/1 min	72 °C/1 min	72 °C/1 min
Final extension	72 °C/7 min	72 °C/7 min	72 °C/7 min

**Table 3 microorganisms-11-02067-t003:** Comparison of colony morphology for *Talaromyces purpureogenus* strains isolated from bee bread and species characterization.

Strain	Diameter on CYA (mm)	Diameter on MEA (mm)	Soluble Pigment on CYA	Colony Texture on MEA	Acid Production on CREA	Exudates on MEA
*Talaromyces* sp. B11	21–25	33–37	Weak red	Velvety to floccose	Absent	Yellow
*Talaromyces* sp. B13	25–26	25–30	Red	Floccose, wrinkled	Absent	Absent
*Talaromyces* sp. B18	23–27	35–37	Red	Velvety, floccose	Absent	Orange
*Talaromyces* sp. B30	21–24	33–40	Absent	Floccose	Absent	Absent
*Talaromyces* sp. B49	25–30	33–36	Absent	Velvety, floccose	Weak	Colorless
*Talaromyces* sp. B69	24–28	26–36	Absent	Velvety	Absent	Absent
*Talaromyces* sp. B195	26–29	27–34	Red	Floccose, funiculous	Absent	Red
*T. purpureogenus* [54]	20–25	30–45	Red	Velvety, floccose	Absent	–

**Table 4 microorganisms-11-02067-t004:** Presence of *Talaromyces* strains in honey bee bread collected from different sampling sites (apiaries) from 2019 to 2020. The values represent the number of hives where *Talaromyces* strains were found relative to the total amount of hives from which the bee bread was collected.

Apiary	Area	April 2019	July 2019	April 2020	July 2020
**KM Zirovnice**	Rural	1/6	0/6	1/6	0/6
**KM agro**	Rural	0/10	0/10	0/10	0/10
**CB Kroclov**	Periphery	No data	No data	1/3	0/3
**CB campus**	Urban	No data	No data	2/4	0/4
**CB Litvinovice**	Suburban	No data	No data	0/4	0/4

**Table 5 microorganisms-11-02067-t005:** Five sequences with the highest identity in the ITS region based on BLAST analysis. The ITS sequence was identical in all seven strains. QC = query cover, I = identity.

ITS Region Seq.	Database Hits	Score	QC/I [%]	Accession
*Talaromyces* strains, 577 bp	*T. purpureogenus* CBS 286.36	1000	97/98.93	NR_121529.1
	*T. rufus* CBS 141834	953	100/96.54	NR_170773.1
	*T. thailandensis* CBS 133147	942	100/96.19	NR_147428.1
	*T. zhenhaiensis* CGMCC 3.16102	937	96/96.96	NR_177565.1
	*T. aspriconidius* CBS 141835	935	97/96.64	NR_170774.1

**Table 6 microorganisms-11-02067-t006:** Minimal inhibitory concentrations of crude methanol extracts from *T. purpureogenus* strains tested at 0.001–2 mg/mL against *B. subtilis*, *P. lautus,* and *P. lactis*, and at 0.39–6.25 mg/mL against *P. alvei* (* using different protocol).

Strains	*B. subtilis*	*P. alvei **	*P. lautus*	*P. lactis*
B11	>2.00	6.25	>2.00	2.00
B13	>2.00	>2.00	>2.00	>2.00
B18	>2.00	0.39	0.50	0.50
B30	>2.00	>2.00	>2.00	2.00
B49	>2.00	>2.00	>2.00	>2.00
B69	>2.00	3.13	2.00	>2.00
B195	2.00	0.39	2.00	2.00

**Table 7 microorganisms-11-02067-t007:** Summary of the dereplication of active fractions derived from the methanol extract of the *T. purpureogenus* strain B18. The structures are shown in Figure 5.

Fraction	*m*/*z*	Adduct	Formula	Name	Structure
26–28	245.1283	[M+H]^+^	C_14_H_16_N_2_O_2_	cyclo-(Phenylalanyl-Prolyl)	1
	211.1438	[M+H]^+^	C_11_H_18_N_2_O_2_	cyclo-(Leucyl-Prolyl)	2
30–31	245.1283	[M+H]^+^	C_14_H_16_N_2_O_2_	cyclo-(Phenylalanyl-Prolyl)	1
	543.2176	[M+H]^+^	C_24_H_34_N_2_O_12_	unknown	
33–34	543.2176	[M+H]^+^	C_24_H_34_N_2_O_12_	unknown	
	191.0334	[M-H_2_O+H]^+^	C_10_H_8_O_5_	5,6,8-trihydroxy-3-methyl-2-benzopyran-1-one	3
39	516.2204	[M+H]^+^	C_22_H_33_N_3_O_11_	unknown	
	436.5186	[M+2H]^2+^	C_39_H_62_N_6_O_16_	l-ornithine, *N^2^*-acetyl-*N^5^*-hydroxy-*N^5^*-(5-hydroxy-3-methyl-1-oxo-2-pentenyl)-, trimol. ester, (Z,Z,Z)-(9CI)	4
114	379.3368	[M+H]^+^	C_28_H_42_	unassigned	
	281.2482	[M+H]^+^	C_18_H_32_O_2_	unassigned	

## Data Availability

Not applicable.

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
