# Peer review of "Extracts of Talaromyces purpureogenus Strains from Apis mellifera Bee Bread Inhibit the Growth of Paenibacillus spp. In Vitro"

_microorganisms, 2023, doi:10.3390/microorganisms11082067_

Round 1
Reviewer 1 Report
The study assesses the interaction between honeybee-derived Talaromyces against Paenibacillus and Aspergillus. I appreciate the originality of the study. The results are properly presented. I suggest some revisions to the methodology part, as some parts are not very clear, and is quite hard to realize how everything was conducted.
Row 99-101 - It is not clear if all fungal species from the primary culture (a mixed culture I suppose) were further isolated and identified; how were the authors able to select Talaromyces spp. only, for the further experiments?
Table S2 shows colony images. The name of the culture media should be presented.
Row 104-107 - explain the acronym MP medium and the producers for all the culture media.
Row 115 - the liquid phase was processed after centrifugation?
Chapter 2.1 presents isolation of Talaromyces from Czech Republic in 2019, while chapter 2.4 presents Germany. It is not clear. If the chaper 2.4 refers to the sites for Talaromyces isolation, I would move it in the first position (before 2.1)
Row 209 - which is the source of Aspergillus species?
"The fungi were cultivated on PDA (VWR International, USA) for 7 days" - ehich fungi? Talaromyces or Aspergillus?
Roe 210 - how was the inoculum prepared?
Row 213 - which extracts? Chapter 2.2 presents two extraction methods: one with formic acid + ultrasound, and one in MP medium.
Chapter 2.6.2 and 2.6.3 - the microdilution method should be presented clearer. The pipetted volumes are also missing.
Author Response
We would like to thank the reviewer for thorough revision and very good points that helped us to improve the clarity of the methods.
We considered and corrected all the points accordingly. Please, see the comments to each reviewer’s point below:
1. Row 99-101 - It is not clear if all fungal species from the primary culture (a mixed culture I suppose) were further isolated and identified; how were the authors able to select Talaromyces spp. only, for the further experiments?
The fungal isolates were purified and identified based on the morphology and ITS sequencing. Only the Talaromyces strains were further used for the analyses. We added the information to make this clear.
2. Table S2 shows colony images. The name of the culture media should be presented.
The description of the cultivation conditions was added.
3. Row 104-107 - explain the acronym MP medium and the producers for all the culture media.
The corrections were performed accordingly.
4. Row 115 - the liquid phase was processed after centrifugation?
We followed a method described by Samson et al. 2010 (Samson, R.A.; Houbraken, J.; Thrane, U.; Frisvad, J.C.; Andersen, B. Food and Indoor Fungi; 2nd ed.; CBS-KNAW Fungal Biodiversity Centre: Utrecht, 2010; ISBN 9789491751189.), where the centrifugation was performed before the HPLC.
5. Chapter 2.1 presents isolation of Talaromyces from Czech Republic in 2019, while chapter 2.4 presents Germany. It is not clear. If the chaper 2.4 refers to the sites for Talaromyces isolation, I would move it in the first position (before 2.1)
Both collections were performed in Czech Republic. Thank you for pointing this out, this mistake must have emerged in the last steps of manuscript editing; it was corrected.
6. Row 209 - which is the source of Aspergillus species?
More information to the strains source was added.
7. "The fungi were cultivated on PDA (VWR International, USA) for 7 days" - ehich fungi? Talaromyces or Aspergillus?
Aspergillus spp. The text was corrected to clarify this.
8. Roe 210 - how was the inoculum prepared?
The information was added.
9. Row 213 - which extracts? Chapter 2.2 presents two extraction methods: one with formic acid + ultrasound, and one in MP medium.
We included the information to the text with the reference to the workflow scheme (Figure 1) that was created for visualization and better understanding of the procedure.
10. Chapter 2.6.2 and 2.6.3 - the microdilution method should be presented clearer. The pipetted volumes are also missing.
Thank you for this point, we added more details to these parts, such as the pipetting volumes.
Reviewer 2 Report
The experiment is soundly designed and manuscript is well written. For section "3.4 Antimicrobial activity and bioactivity-guided dereplication" at line 325 or Table 6, I think authors should provide results from all the treatment levels or strains even results are proved to be less effective. This section is the key information for this research. Line 40, please delete "As". I recommend acceptance after considering this minor change.
Author Response
We appreciate the positive feedback of the reviewer and the suggested corrections. Please, see the authors’ comments to the suggested changes.
1. For section "3.4 Antimicrobial activity and bioactivity-guided dereplication" at line 325 or Table 6, I think authors should provide results from all the treatment levels or strains even results are proved to be less effective.
Thank you for pointing this out. In the main text, we presented the results for all the tested strains, where at least one strain showed an effect. All the results are further summarized in the supplementary material.
2. Line 40, please delete "As".
We performed the suggested correction.
Reviewer 3 Report
This manuscript provides the antimicrobial activity of Talaromyces purpureogenus strains isolated from bee bread against Paenibacillus alvei in vitro.
There are some places may need to be revise as below:
1 Tittle. I think it should be “Methanol Extracts of Talaromyces purpureogenus strains isolated from Apis mellifera bee bread inhibit the growth of Paenibacillus spp. in vitro” or other expressions, which shows that extracts but not Talaromyces purpureogenus strains has antibacterial activity.
2 Lines 25, 74, 141, 204, 264 etc. These sentences should be passive.
3 Line 33. Please provide the full name of Talaromyces purpureogenus.
4 Line 55. Presented?
5 Line 99. Add “(grade, com.)” of “Tween-80”. These informations of Czapek yeast autolysate (CYA) medium, malt extract agar (MEA), yeast extract supplemented (YES) medium, creatine sucrose agar (CREA) and others are also necessary.
6 Line 103. “Agr” should be PDA?
7 Line 115. The frequency of ultrasonic trt.
8 Line 146. Revise “cycling” as procedure?
9 Line 152. Change “Parameters that vary between targets are highlighted in bold.” as Note.
10 Line 157. Add “, com.” of BLAST. Other software also need the version, company or website.
11 Lines 173-178. I think these informations should be combined with 2.1.
12 Line 248. Please add 2.7 Data analysis.
13 Line 261. Remove [54].
14 Line 381. Delete “The structures are shown in Figure 5.”
And some infromation of 1to 4, such as 1: cyclo-(Phenylalanyl-Prolyl) …
15 Part 5 Conclusions. I think this part should focus on the results of T. purpureogeneus in this experiment.

The same as "Comments and Suggestions" for authors.
1 Tittle. I think it should be “Methanol Extracts of Talaromyces purpureogenus strains isolated from Apis mellifera bee bread inhibit the growth of Paenibacillus spp. in vitro” or other expressions, which shows that extracts but not Talaromyces purpureogenus strains has antibacterial activity.
2 Lines 25, 74, 141, 204, 264 etc. These sentences should be passive.
3 Line 33. Please provide the full name of Talaromyces purpureogenus.
4 Line 55. Presented?
5 Line 99. Add “(grade, com.)” of “Tween-80”. These informations of Czapek yeast autolysate (CYA) medium, malt extract agar (MEA), yeast extract supplemented (YES) medium, creatine sucrose agar (CREA) and others are also necessary.
6 Line 103. “Agr” should be PDA?
7 Line 115. The frequency of ultrasonic trt.
8 Line 146. Revise “cycling” as procedure?
9 Line 152. Change “Parameters that vary between targets are highlighted in bold.” as Note.
10 Line 157. Add “, com.” of BLAST. Other software also need the version, company or website.
11 Lines 173-178. I think these informations should be combined with 2.1.
12 Line 248. Please add 2.7 Data analysis.
13 Line 261. Remove [54].
14 Line 381. Delete “The structures are shown in Figure 5.”
And some infromation of 1to 4, such as 1: cyclo-(Phenylalanyl-Prolyl) …
15 Part 5 Conclusions. I think this part should focus on the results of T. purpureogeneus in this experiment.
Author Response
The authors thank the reviewer for the provided recommendations that helped us to improve the clarity and readability of the text. We considered all the points and performed the corrections accordingly. Please, see the point-by-point response:
1 Tittle. I think it should be “Methanol Extracts of Talaromyces purpureogenus strains isolated from Apis mellifera bee bread inhibit the growth of Paenibacillus spp. in vitro” or other expressions, which shows that extracts but not Talaromyces purpureogenus strains has antibacterial activity.
Thank you for your recommendation. To make the title more accurate, we changed it to: "Extracts of Talaromyces purpureogenus strains isolated from Apis mellifera bee bread inhibit the growth of Paenibacillus spp. in vitro”
2 Lines 25, 74, 141, 204, 264 etc. These sentences should be passive.
Thank you for the comment. We used the passive and active intentionally for an easy recognition of our results and the results from the literature as well as better readability of the text. The manuscript was checked by a professional editor; therefore, we assume the style fulfills good academic writing standards.
3 Line 33. Please provide the full name of Talaromyces purpureogenus.
We agree that the study is particularly related to the T. purpureogenus, however, given that we found another species of Talaromyces in the A. mellifera bee bread, T. piceae, we think the keyword of the genus is suitable.
4 Line 55. Presented?
The word “present” was used as an adjective here, not a verb in past tense form.
5 Line 99. Add “(grade, com.)” of “Tween-80”. These informations of Czapek yeast autolysate (CYA) medium, malt extract agar (MEA), yeast extract supplemented (YES) medium, creatine sucrose agar (CREA) and others are also necessary.
We added the information to the chemicals. For the characterization, we followed the method by Samson et al. 2010 (cited), where also the producers are recommended.
6 Line 103. “Agr” should be PDA?
Thank you for the question. The solution for the spore suspension preparation contains low concentration of agar.
7 Line 115. The frequency of ultrasonic trt.
Thank you for the comment. We added the frequency.
8 Line 146. Revise “cycling” as procedure?
We changed the “cycling” conditions to “reaction” conditions.
9 Line 152. Change “Parameters that vary between targets are highlighted in bold.” as Note.
We performed the change according to the suggestion.
10 Line 157. Add “, com.” of BLAST. Other software also need the version, company or website.
We added the reference to BLAST and version of the software to gimp. Unfortunately, we were not able to further find the version of the BLAST database.
11 Lines 173-178. I think these informations should be combined with 2.1.
We appreciate and understand the comment. Nevertheless, in the chapter 2.1, there is information related to the strains that were used for the further analyses. The information in the mentioned lines is related to isolation of the strains that were characterized but not further analyzed for the biological activity etc. We assume moving this part to 2.1 might be therefore confusing.
12 Line 248. Please add 2.7 Data analysis.
Because of the complexity of our analyses (for example the metabolic fingerprinting), we assumed it is better to include the information of the data analysis to the related parts of the study.
13 Line 261. Remove [54].
The correction was performed
14 Line 381. Delete “The structures are shown in Figure 5.”
And some infromation of 1to 4, such as 1: cyclo-(Phenylalanyl-Prolyl) …
We added the information on the structure description, according to the reviewer’s recommendation.
Unfortunately, it was not possible to include the structures to the Table 7 due to the space limitations. Therefore, we added the Figure 5 and referred to it in the table as we considered this as the most understandable way to present the results.
15 Part 5 Conclusions. I think this part should focus on the results of T. purpureogeneus in this experiment.
Thank you for the suggestion, the conclusion involves rather the outcomes than the results that are discussed before.